**Subject Category:**
Biology (whole organism)

biomedical engineering/bioengineering

vibration stimulation, electromyography, fatigue, neuromuscular response, isometric contraction

**Author for correspondence:**
Amit N. Pujari
e-mail: amit.pujari@ieee.org

# Fatiguing effects of indirect vibration stimulation in upper limb muscles: pre, post and during isometric contractions superimposed on upper limb vibration

Amit N. Pujari[1,2], Richard D. Neilson[1]
and Marco Cardinale[3]

[1]School of Engineering, University of Aberdeen, Aberdeen AB24 3DX, UK
[2]School of Engineering and Technology, University of Hertfordshire, Hatfield AL10 9AB, UK
[3]Department of Computer Science, University College London, London WC1E 6EA, UK

ANP, 0000-0003-1688-4448; MC, 0000-0002-2777-8707

Whole-body vibration and upper limb vibration (ULV) continue to gain popularity as exercise intervention for rehabilitation and sports applications. However, the fatiguing effects of indirect vibration stimulation are not yet fully understood. We investigated the effects of ULV stimulation superimposed on fatiguing isometric contractions using a purpose developed upper limb stimulation device. Thirteen healthy volunteers were exposed to both ULV superimposed to fatiguing isometric contractions (V) and isometric contractions alone Control (C). Both Vibration (V) and Control (C) exercises were performed at 80% of the maximum voluntary contractions. The stimulation used was 30 Hz frequency of 0.4 mm amplitude. Surface-electromyographic (EMG) activity of the Biceps Brachii, Triceps Brachii and Flexor Carpi Radialis were measured. EMG amplitude (EMGrms) and mean frequency (MEF) were computed to quantify muscle activity and fatigue levels. All muscles displayed significantly higher reduction in MEFs and a corresponding significant increase in EMGrms with the V than the Control, during fatiguing contractions ($p < 0.05$). Post vibration, all muscles showed higher levels of MEFs after recovery compared to the control. Our results show that near-maximal isometric fatiguing contractions superimposed on vibration stimulation lead to a higher rate of fatigue development compared to the isometric contraction alone in the upper limb muscles. Results also show higher manifestation of mechanical fatigue post treatment with

vibration compared to the control. Vibration superimposed on isometric contraction not only seems to alter the neuromuscular function during fatiguing efforts by inducing higher neuromuscular load but also post vibration treatment.

## 1. Introduction

Vibration stimulation has been used as a diagnostic tool in neurological studies since the 1970s [1]. The actual delivery of vibration stimulation can be broadly classified into two categories: (1) stimulation directly applied to a muscle or tendon and (2) indirect vibration delivered to the limbs via the bones, joints and ligaments. Indirect vibration delivered to the upper limbs is referred to as upper limb vibration (ULV) or local vibration and when delivered to the lower limbs, it is commonly referred as whole-body vibration (WBV). In recent years, this so-called 'indirect vibration' has been increasingly investigated for its positive effects. Various studies have proposed the use of vibration stimulation for increasing muscle strength, muscle power, balance and bone remodelling [2–6]. Consequently, given the potential benefits of vibration stimulation, it has been suggested that this therapeutic modality could represent a viable intervention to improve sport performance and/or enhance rehabilitation from injury [7,8].

While the potential beneficial effects of vibration stimulation on muscle and bone form and functions are recognized now in various populations [9,10], a lack of consensus exists on the biological mechanisms responsible for such adaptations. In particular, when assessing the typical neuromuscular responses observed during vibration stimulation, a muscle tuning hypothesis was suggested [11]. The main idea behind this approach was that the increase in neuromuscular activity observed during vibration is a strategy of working muscles to damp the vibratory stimulus [12] and such activity is modulated by sensory receptors [13]. Exposure to such stimulation can therefore enhance neuromuscular function due to the stimulation of sensory motor pathways, leading to an increase in force generating capacity in skeletal muscle.

In this regard, direct vibration stimulation has been shown to enhance muscle spindle activity resulting in excitatory response of the primary and secondary endings [1,14], the excitatory response being known as tonic vibration reflex (TVR) [15,16]. It has also been observed that the TVR response is influenced by the vibration location, the initial length of muscle, i.e. pre-stretch and the vibration frequency and amplitude [17,18]. Considering the effect of muscle length, contraction and vibration frequency as well as amplitude in grading the neuromuscular responses to vibration stimulation, one way to further improve the effectiveness of the vibration exercise could be the superimposition of isometric exercises on vibration stimulation.

Therefore, to control and exploit the effect of changes in muscle length, contraction, vibration frequency and amplitude can have on the final neuromuscular response, we have recently developed a portable vibration stimulation device for the upper limbs [19]. This novel and portable device enables the user to perform isometric contraction exercise(s) of various intensities and superimpose a vibratory stimulus. The device also allows precise and independent control over the delivery of vibration frequency and amplitude; a feature none of the existing devices can offer [19]. Further, to the best of the authors' knowledge, only a few studies have investigated the effects of graded isometric contractions superimposed on vibration stimulation in the upper limbs [20]. However, these studies did not analyse the fatiguing effects of vibration both during and after vibration stimulation, post exercise. The device used in these studies and developed by Mischi *et al*. [20–22] is capable of superimposing vibration to various levels of muscle contraction by means of a pulley system with an operating frequency of 0–60 Hz with a limited pulling force. With this approach, it was shown that the bicep and tricep muscles show an increased EMG activity when vibration is superimposed to force production [22] and that this modality of exercise determines a higher degree of neuromuscular fatigue [20].

However, considering the size of Mischi *et al*.'s equipment and its limited operating envelope, it would be beneficial to find a technical solution to provide vibration exercise with smaller equipment and a wider operating envelope. For this reason, and to overcome the limitations of other devices [19], we aimed to develop a smaller and portable vibratory exercise device for the upper limbs and assess its feasibility as a means to increase neuromuscular performance in healthy individuals.

Furthermore, we aimed to assess the effects of vibratory stimulation with this novel device on a higher level of muscle tension to determine the typical neuromuscular responses and identify the possibility to induce fatigue, in the target muscles to a larger degree, than muscle tension alone. Typically, myoelectric manifestations of fatigue are said to be characterized by the two main physiological factors, peripheral (muscle) fatigue and central fatigue [23]. Mean frequency (MEF) of

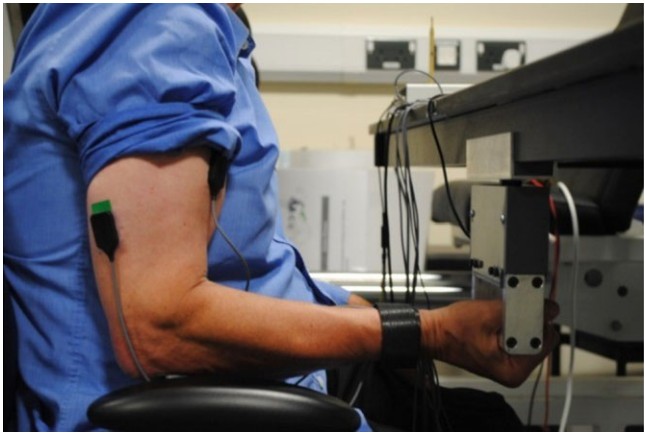

**Figure 1.** Photograph showing volunteer posture and sensor placement used for the tests on the ULV device.

surface electromyography (sEMG) has been suggested to be a strong indicator of the peripheral fatigue [22,24]. Based on the above premise, in this study we investigated the behaviour of MEF values of sEMG as a representation of myoelectric manifestations of fatigue.

Considering the capability of this novel device to deliver precise vibratory stimuli on a wider envelope of muscle contraction [19], we hypothesized that when compared to the Control (C) condition:

(i) The Vibration stimulation (V) delivered through this novel device would induce an increased neuromuscular activity in the Biceps Brachii (BB), Triceps Brachii (TB) and Flexor Carpi Radialis (Forearm FCR) muscles as measured by surface EMG.
(ii) The V stimulation delivered through this novel device would induce a higher degree of neuromuscular fatigue in these muscles, compared to C.

# 2. Methods

## 2.1. Participants

Seven female and six male (Age 28 years ± 7.24, Height 173 cm ± 13.04, Weight 73.16 kg ± 11.19) healthy volunteer participants were recruited through the University of Aberdeen, Biomedical Engineering laboratory. The level of physical training of the participants varied from sedentary to amateur athlete. Written and informed consent was given by each volunteer. Experiments were approved by the College Ethics Review Board (CERB) of the College of Life Sciences and Medicine, University of Aberdeen, Aberdeen, Scotland, UK.

Exclusion criteria included history of heart disease and thrombosis, recent musculoskeletal injury, recent fractures in the upper limbs, metallic plates in the bones of the upper limbs, pacemakers, and circulatory diseases.

## 2.2. Experimental set-up

Trials were performed with the set-up previously reported in detail [19,25].

Briefly, the participants sat on the chair with back and arm rest with their arm flexed at a 90° elbow angle while pushing against a vibrating hand grip (figure 1). Participants gripped the handle bar of the ULV device tightly while exercising. Each participant received detailed instructions on the posture and relevant exercise protocol. Throughout the exercise, it was ensured that the participants' back was supported by the chair back rest and their shoulders consistently aligned in transverse plane. It was ensured that there was no contact between participants' elbows and the chair handle (arm rest).

## 2.3. Study design

Participants were asked to visit the laboratory three times (first visit to establish the Maximum Voluntary Contraction (MVC) and familiarize them with the equipment and next two visits to receive the experimental treatment, see protocol in figure 2).

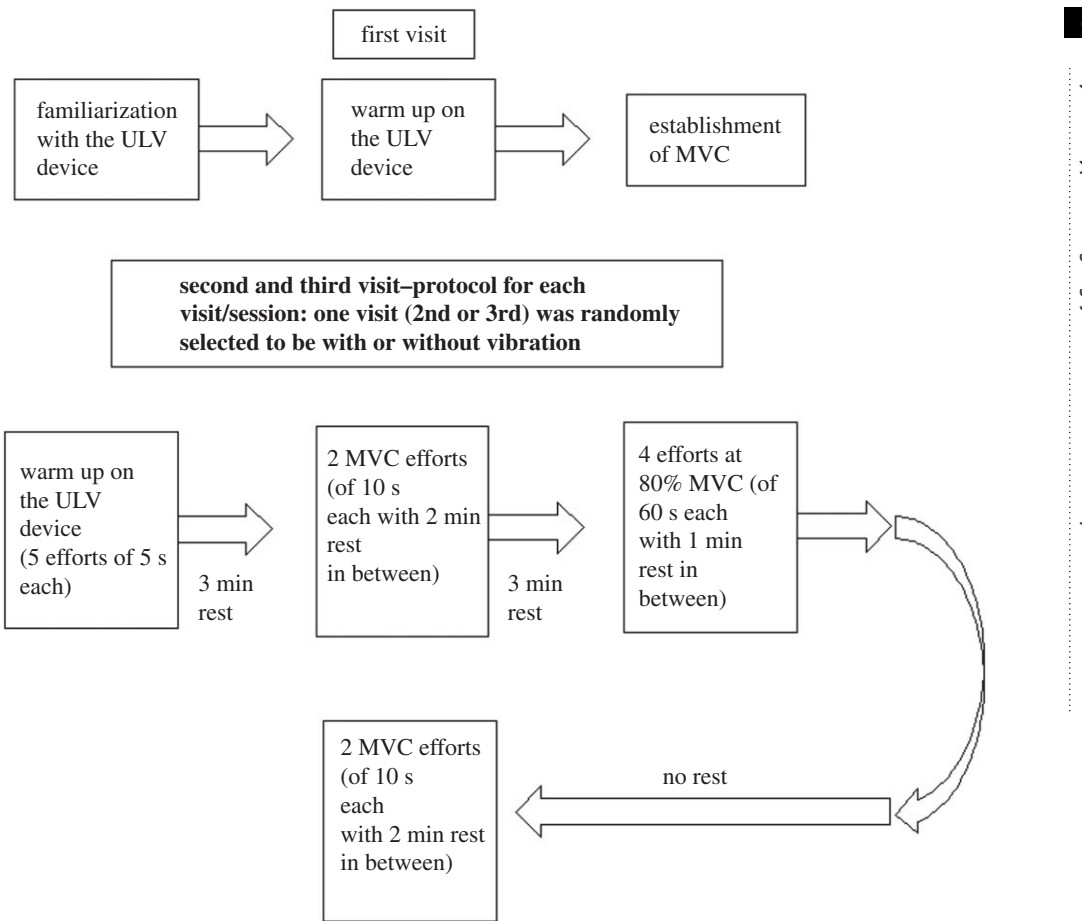

**Figure 2.** Schematic describing the research protocol of the experiments.

In each laboratory visit, the participants were fitted with the three-surface electromyography (sEMG) electrodes placed on the upper limb muscles: Biceps Brachii (BB), Triceps Brachii (TB) and Flexor Carpi Radialis (Forearm-FCR) muscle.

In the first visit, each participant was introduced to the upper limb vibration (ULV) device (figures 1 and 2). Then the participant performed isometric arm flexion exercise of various intensities as a warm-up by holding on to the hand grip of the ULV device with elbow flexed at 90°. After the initial warm-up and familiarization, the participants were asked to perform five Maximal Voluntary Contractions (MVC) performing an arm curl maximal effort for 10 s, each separated by a 5 min period of no activity/ resting. The largest of the five values was used as the MVC value for that participant.

In the second and third visits, the participants went through the following vibration exercise treatments with a randomized cross-over design.

(A) Treatment 1: four sets of 1 min muscle contraction (referred to as Control Fatiguing Contractions, i.e. CFC1, CFC2, CFC3 and CFC4) with 80% of MVC.
(B) Treatment 2: four sets of 1 min muscle contraction (referred to as Vibration Fatiguing Contractions, i.e. VFC1, VFC2, VFC3 and VFC4) with 80% of MVC coupled with 30 Hz vibration at 0.4 mm amplitude.

Treatments 1 and 2 consisted of an isometric arm curl (flexion) against the ULV device hand grip with the elbow flexed at 90° at the target force, for 60 s. Eight measurements were carried out per treatment session (2 MVC efforts + 4 efforts at 80% MVC + 2 MVC efforts) with treatment 1 or 2 allocated randomly (figure 2). Thus, a total of 16 measurements taken in the two treatment sessions (second and third sessions).

Visual real-time feedback from a load cell in the device and verbal feedback from the device operator were used to achieve the 80% of MVC force level. During each measurement, the sEMG activity of the designated muscles was recorded and stored for analysis along with the vibration characteristics and

the force production details (load cell values). The vibration being delivered was continuously monitored with an accelerometer (ADXL-330, Analog Devices, MA, USA; ADXL-330).

All the procedures were non-invasive. Medical grade, single-use sEMG electrodes were used. Participants wore appropriate clothing to facilitate sensor placement on the upper limbs. Randomized cross-over design was used to carry out these 16 exercise treatments/tests.

## 2.4. Instructions to the participants

Participants were asked and guided to maintain consistency in their hand grip positions and elbow angles. Throughout the tests, both the hand grip position and the elbow angle were measured with a ruler and a goniometer, respectively. Participants received both verbal and visual (real-time graphical values on the PC) feedback to assist them in maintaining a constant force level.

## 2.5. Fatigue and safety

A minimum of 72 h of recovery time was allowed between any two testing sessions to avoid any residue of delayed onset of muscle soreness and fatigue. Also, a general log of the participants' daily physical activity excluding the vibration tests was kept, i.e. participants who undertook any form of regular/irregular physical exercise, e.g. strength or resistance training, etc. At least a 1-day time gap was allowed between the regular physical exercise and ULV exercise, to avoid any effect of muscle fatigue (especially of upper limbs).

## 2.6. EMG measurements and processing

sEMG was recorded from the BB, TB and FCR during all exercise conditions according to recommendations reported in the literature, widely known as SENIAM guidelines [26]. Active bipolar electrodes (DelSys, Inc.; model: DE 2.1) were aligned with the muscle fibre direction and placed between the tendon and the muscle belly. Precise location of the electrode placement used on each muscle can be found on the SENIAM website [27]. To minimize the impedance and to ensure a proper contact, the skin was shaved as necessary, lightly abraded and cleaned with 70% isopropyl alcohol. The reference electrode was placed on an electrically inactive area of the lumbar spine (the anterior superior iliac spine). The sEMG electrodes and cables were secured to the subject's skin with medical tape. Active grounding and shielding of the cables was carried out to minimize electromagnetic inference [28]. The sEMG signals were sampled at 1000 Hz, amplified with a gain of 1000 and analogue filtered for a 20–450 Hz band pass with DelSys hardware (DelSys, Inc.; model: Bagnoli-4). It is important to note that the signal was band pass filtered (20–450 Hz) at the point of acquisition to remove any low-frequency components. Data acquisition was performed through a 16 bit data acquisition card (National Instruments Corp.; model: PCI-6220 M) and EMGWorks (DelSys, Inc.) software.

Subsequent data processing and analysis was performed with custom written MATLAB code (The Mathworks, Inc.; v. 8) routine. Any baseline offset of the sEMG data was removed by subtracting the mean.

The root-mean-square (RMS, i.e. EMGrms) was used to estimate the neuromuscular activation. The RMS was calculated using the moving window technique. Initially, the RMS was calculated for each window, and then the RMS for the entire data length was obtained by averaging the individual RMS values of each window.

The Mean Frequency (MEF) and Median Frequency (MDF) of the sEMG data were also obtained. These spectral estimators were also derived by moving window technique. Due to the similarity of trends shown by MEF and MDF, MDF results are excluded from any further discussion in this paper.

For both the amplitude and spectral estimation, the hamming window with a length of 1 s and no overlap was used. It has been shown that the choice of the window does not have a critical bearing on the spectral estimators like MEF and Power Spectral Density (PSD) [29]. Furthermore, for isometric, constant force and fatiguing contractions, the signal is regarded as stationary for epoch/window duration of 1–2 s. Previous studies suggest that epoch duration between 500 ms and 1 s provide better spectral estimation [29–31]. Also, it has been shown that window overlapping does not provide any significant benefits [29]. Based on these recommendations, the window length was kept to 1 s without any overlap.

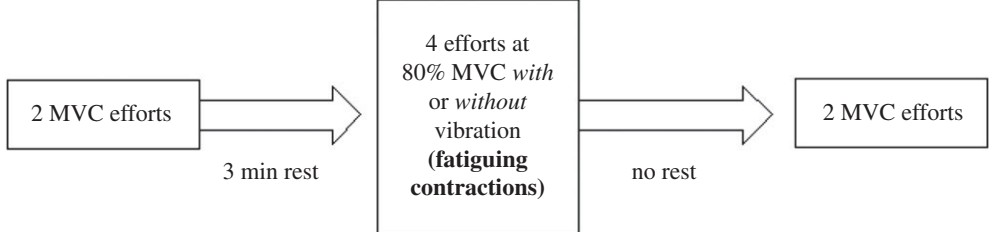

**Figure 3.** Schematic describing the MVC and fatiguing contractions' order.

**Table 1.** Average times (in seconds) completed by participants during fatiguing contractions and their standard deviations.

| control | CFC1 | CFC2 | CFC3 | CFC4 |
|---|---|---|---|---|
| times (s) | 53 | 51.30 | 44.53 | 40.53 |
| s.d. | 8.87 | 11.96 | 14.17 | 17.03 |
| vibration | VFC1 | VFC2 | VFC3 | VFC4 |
| times (s) | 49.92 | 49.23 | 43.50 | 40.53 |
| s.d. | 13.18 | 12.79 | 14.83 | 16.95 |

The primary objective of this study was to quantify the fatiguing effect of ULV stimulation superimposed on isometric contraction on the targeted muscles. Due to this objective, EMG data of the fatiguing contractions (figures 2 and 3) was processed in a specific manner. These processing techniques are detailed below.

## 2.7. EMG data processing of the fatiguing contractions

The target was to maintain the 80% of MVC force for 60 s under both control fatiguing contractions (CFCs) and vibration fatiguing contractions (VFCs). However, while some participants managed to maintain the target force for 60 s, others did not. As the treatment progressed from the first fatiguing effort to the second and so on (i.e. from CFC1 to CFC4 or from VFC1 to VFC4), the time duration for which an individual could maintain the target force dropped to 30 s in some cases (table 1) by the time the last fatiguing contraction was reached (i.e. CFC4 or VFC4).

To compare the relative fatigue levels attained by the individuals while performing the same relative efforts, the EMG data selection for the analysis began once the target force level of 80% of MVC was achieved and ended when the force level dropped down to 60% of MVC. Thus, the duration for which the EMG data were selected was precisely connected to an individual's ability to sustain the target force. As expected, this period varied with the individual and with the progression to the successive fatiguing contraction (table 1).

The selection of EMG data was then divided into five equal sections (e.g. 60 s selection was divided into five sections of 12 s each). Then an average for each section was calculated to give a single mean value for that section of the fatigue effort. Both the time domain (EMGrms) and frequency domain (mean (MEF) frequency) calculations were performed on the same selection of data. To carry out statistical analysis, each single mean value obtained for a specific section of the fatigue effort (1–5), for a single participant, was compared with the respective mean values of the other participants.

## 2.8. Line artifact removal

Some authors have filtered the peaks in sEMG spectra coinciding with the vibration stimulation frequencies assuming them to be motion artefacts [32]. However, it is still unclear whether the spectral peaks correlating with the stimulation frequencies are in fact motion artefacts [31] or stretch reflexes [13]. Recent evidence suggests that these peaks might indeed be stretch reflexes [33,34]. Considering the present ambiguity about the existence of motion artefacts and increasing evidence of the presence of stretch reflex [13,33,34], only the spectra exhibiting the largest power and hence potential to skew the results were removed. The largest spectra were found to be at 50 Hz coinciding with mains

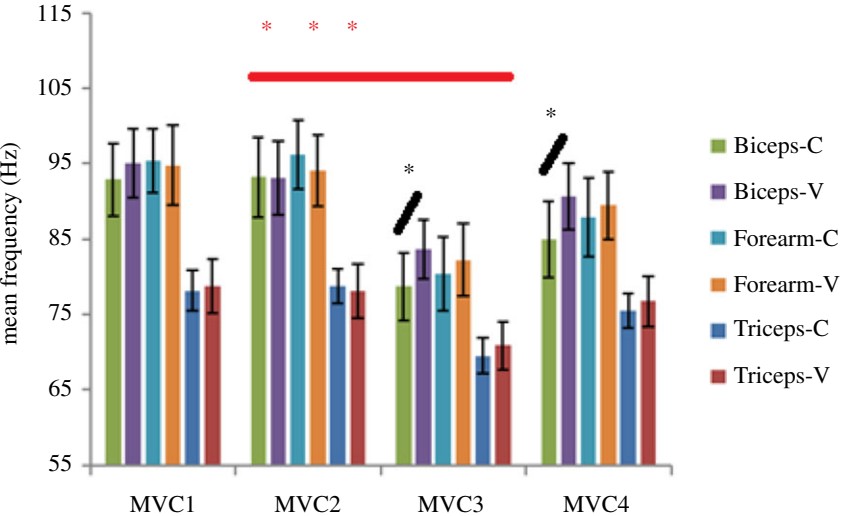

**Figure 4.** Mean frequency (MEF) values for the Biceps, Forearm and Triceps during each MVC effort pre (MVC1 and MVC2) and post (MV3 and MVC4) fatigue exercise under control (no vibration) and vibration (30 Hz–0.4 mm) conditions. Black asterisk represents comparison between MEF values of each control and vibration effort showing statistical significance (*). Red asterisk represents comparison between MEF values of control efforts performed just before (MVC2) and after (MVC3) the fatiguing exercise, showing statistical significance (*). Corresponding vibration comparison (between MVC2 and MVC3 MEF values were found to be non-significant).

frequency of some of the equipment. A Butterworth notch filter (10th order, cut-off frequencies 49.5–50.5 Hz) was employed to remove the components at this frequency.

## 2.9. Statistical analysis

Normalization was performed by dividing the EMGrms of the entire section of the data value to be normalized by the maximum value obtained from the MVC effort of each participant. To identify whether the EMGrms values differ significantly between subsequent fatiguing contractions and between the control and vibration conditions, a two-way ANOVA test was employed. The two interventions (C and V) and four contractions (MVC1 to MVC4) were used to compare between the EMGrms of subsequent fatiguing contractions as well as between the control and each vibration condition one at a time. Alpha was set at 0.05. In each case, a significant difference was defined for a computed $p$-value $\leq 0.05$. Paired student $t$-tests (one tail, different variance) were employed to compare the sEMG responses between the C and V conditions and to establish the significance level ($p$-value) of the deviations from the means. Choice of $t$-tests to establish the significance level was based on Gaussian distribution of the data. Statistical analysis was carried out using the SigmaPlot statistical software package (Systat Software Inc.; Version SigmaPlot 12).

In the results and figures below, statistically significant differences ($p$-value $\leq 0.05$) are noted by asterisk (*).

## 3. Results

Firstly, EMG (frequency and amplitude) results pre and post fatiguing exercise are presented followed by EMG (frequency and amplitude) results during fatiguing exercise.

### 3.1. EMG frequency results—pre and post fatigue exercise—mean frequency (MEF)

MVC efforts post both the vibration and control fatiguing treatments (MVC3s and MVC4s) showed a reduction in MEF values in comparison with their respective pre-treatment effort values (MVC1s and MVC2s), an expected effect of muscle fatigue manifested in myoelectric signal (figure 4). Further, for MVC3 and MVC4, all the post vibration treatment frequency values were higher than the respective post control treatment values, which is unexpected (figure 4). However, only the Biceps' MVC3 and

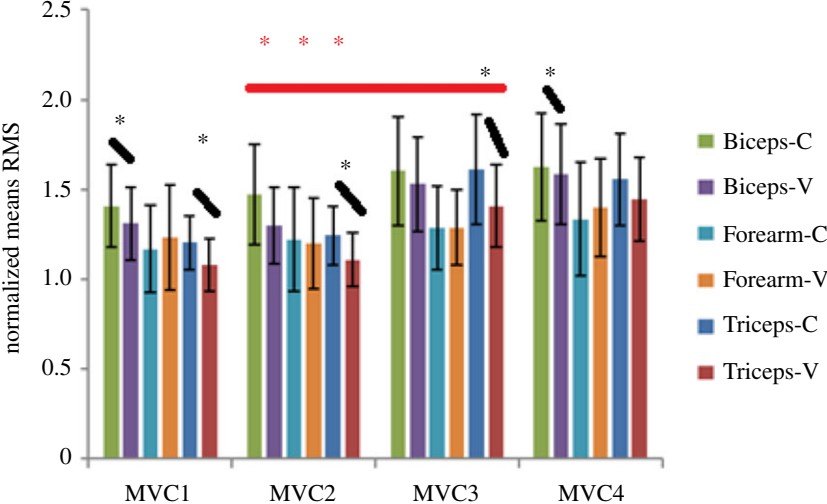

**Figure 5.** Normalized EMGrms values for the Biceps, Forearm and Triceps during each MVC effort pre (MVC1 and MVC2) and post (MV3 and MVC4) fatigue exercise under control (no vibration) and vibration (30 Hz–0.4 mm) conditions. Black asterisk represents comparison between EMGrms values of each control and vibration effort showing statistical significance (*). Red asterisk represents comparison between EMGrms values of control efforts performed just before (MVC2) and after (MVC3) the fatiguing exercise, showing statistical significance (*). Corresponding vibration comparison (between MVC2 and MVC3 EMGrms values were found to be non-significant).

MVC4 showed statistically significant ($p < 0.034$ and $0.029$, respectively), higher values of MEF when the control (C) and vibrations (V) values were compared (figure 4).

Comparison between pre and post fatiguing contraction MVCs for the vibration treatment (vibration MVC2 and MVC3) resulted in statistically non-significant difference, whereas a similar comparison of the control treatment efforts indicates statistically significant ($p < 0.001, 0.001$ and $00.2$ for Biceps, Triceps and Forearm, respectively) differences in all the muscle groups (figure 4). Also, the vibration MEF values for MVC4 for all the muscles groups were higher compared with corresponding control MVC4 values, suggesting continued effect of vibration treatment post exercise period of 3 min (figure 4).

## 3.2. EMG amplitude results—pre and post fatigue exercise—EMGrms

In the post treatment MVC measurements (i.e. MVC3 and MVC4), an increase in mean EMGrms was observed (figure 5). Further, within these post treatment effort EMGrms values, i.e. MVC3 and MVC4, Biceps (MVC3, MVC4) and Triceps (MVC3) showed significant ($p < 0.191$ and $0.371$ for Biceps and $0.045$ for Triceps, respectively) decrease in EMGrms, when C and V were compared (figure 5). Comparison between the pre and post fatiguing contraction MVCs (i.e. between MVC2 and MVC3) of the controls results in statistically significant ($p < 0.024, 0.027$ and $0.025$ for Biceps, Triceps and Forearm, respectively) increases in EMGrms of all the muscles (figure 5). Whereas, comparison between the pre and post fatiguing contractions of vibration MVCs (i.e. between MVC2 and MVC3) result in non-significant differences in all the muscle groups (figure 5).

## 3.3. EMG frequency results—during fatigue exercise—mean frequency (MEF)

Within each fatiguing effort, i.e. CFC or VFC, as the exercise/treatment progressed in time (i.e. consecutive sections of the fatigue tasks), the MEF's trend under vibration conditions separated from that of the control conditions (figures 6–13). The MEFs under vibration condition efforts decreased with a steeper slope indicating a higher rate of fatigue manifestation compared to the corresponding control effort (figures 6–13). The separation between vibration and control mean frequencies is more evident for the Triceps and Forearm muscles than the Biceps (figures 6–13).

As the consecutive fatiguing efforts progressed, i.e. from CFC1, CFC2, CFC3 to CFC4 and for VFC1 to VFC4, the separation between vibration and control MEFs increased (figures 6–13). Thus, MEF values under vibration condition decrease further in value, in comparison with their control counterpart at the last effort (CFC4 versus VFC4), as compared to their separation in the initial efforts (e.g. CFC1

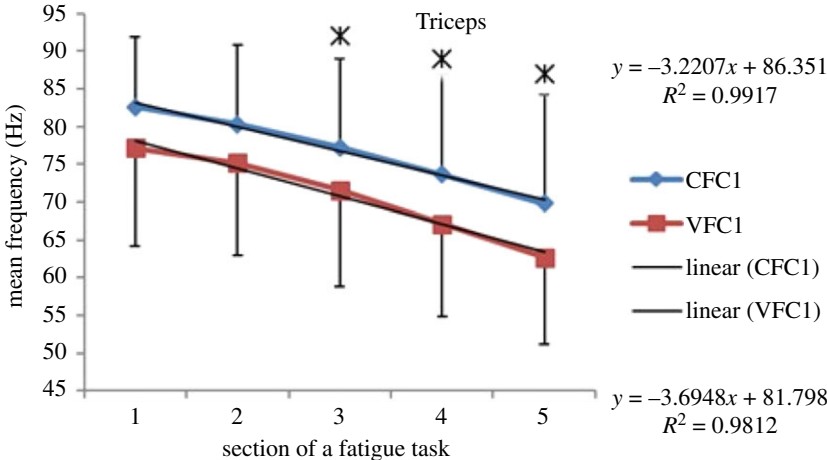

**Figure 6.** MEF values for the Triceps for the five consecutive sections of the fatigue effort during the progression of the first fatiguing exercise effort performed, under control and vibration condition, CFC1 versus VCF1. Y-values represent slopes of the best linear fit for C (top) and V (bottom) conditions quantified with least-squares method.

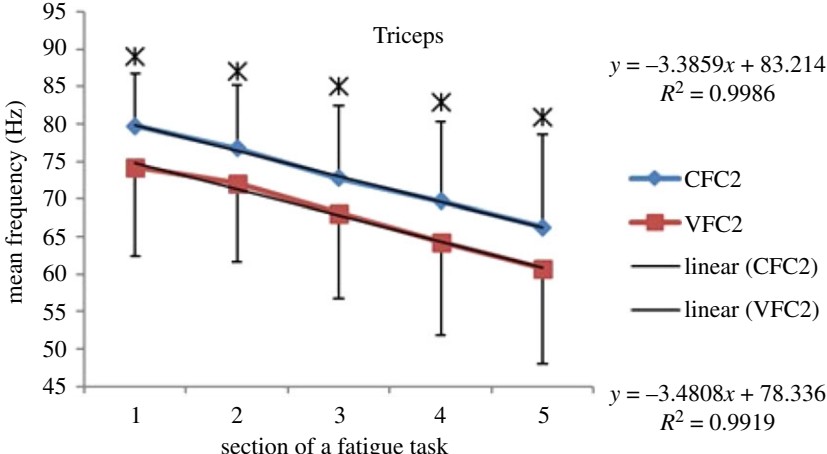

**Figure 7.** MEF values for the Triceps for the five consecutive sections of the fatigue effort during the progression of the second fatiguing exercise effort performed, under control and vibration condition, CFC2 versus VCF2. Y-values represent slopes of the best linear fit for C (top) and V (bottom) conditions quantified with least-squares method.

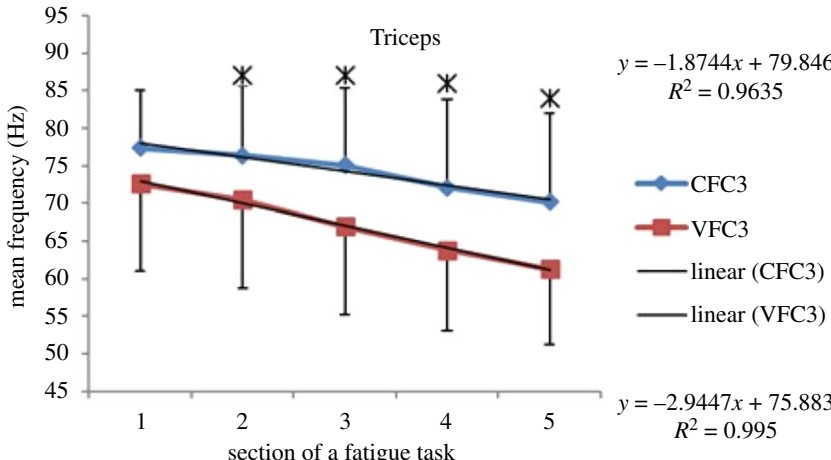

**Figure 8.** MEF values for the Triceps for the five consecutive sections of the fatigue effort during the progression of the third fatiguing exercise effort performed, under control and vibration condition, CFC3 versus VCF3. Y-values represent slopes of the best linear fit for C (top) and V (bottom) conditions quantified with least-squares method.

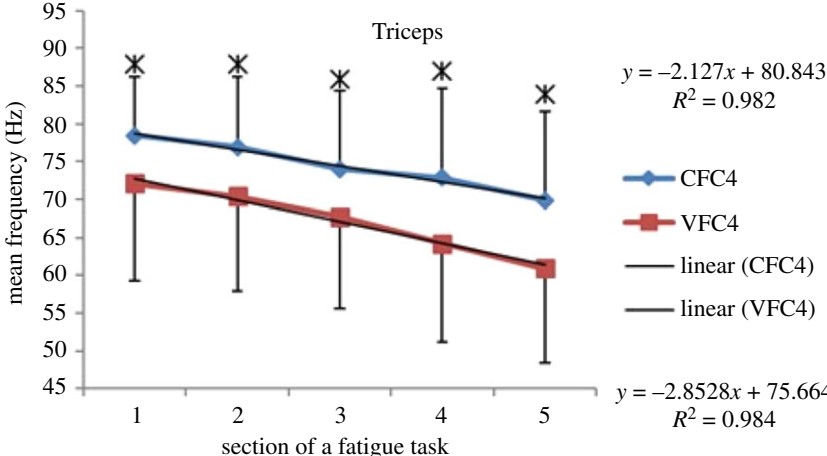

**Figure 9.** MEF values for the Triceps for the five consecutive sections of the fatigue effort during the progression of the fourth fatiguing exercise effort performed, under control and vibration condition, CFC4 versus VCF4. Y-values represent slopes of the best linear fit for C (top) and V (bottom) conditions quantified with least-squares method.

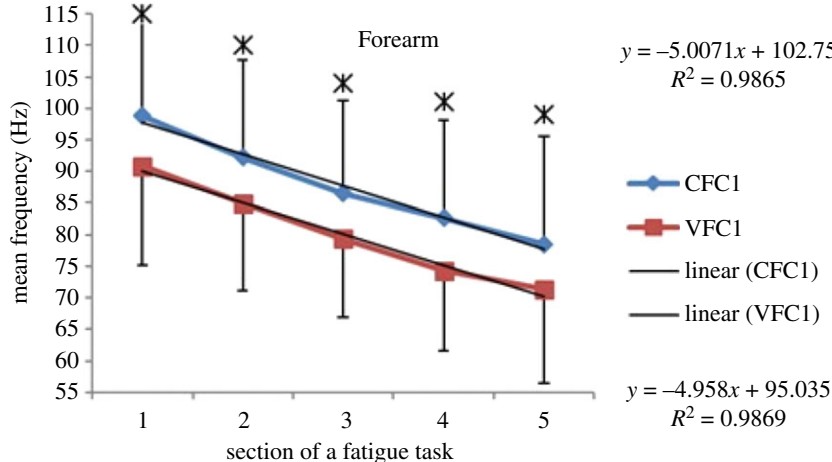

**Figure 10.** MEF values for the Forearm for the five consecutive sections of the fatigue effort during the progression of the first fatiguing exercise effort performed, under control and vibration condition, CFC1 versus VCF1. Y-values represent slopes of the best linear fit for C (top) and V (bottom) conditions quantified with least-squares method.

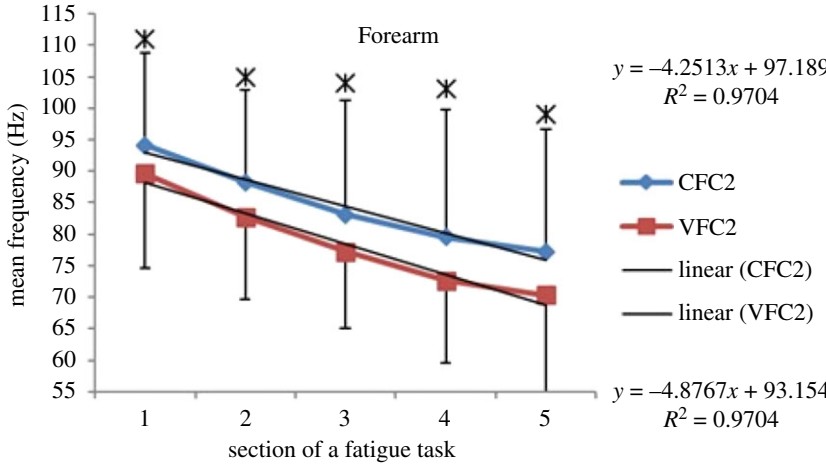

**Figure 11.** MEF values for the Forearm for the five consecutive sections of the fatigue effort during the progression of the second fatiguing exercise effort performed, under control and vibration condition, CFC2 versus VCF2. Y-values represent slopes of the best linear fit for C (top) and V (bottom) conditions respectively quantified with least-squares method.

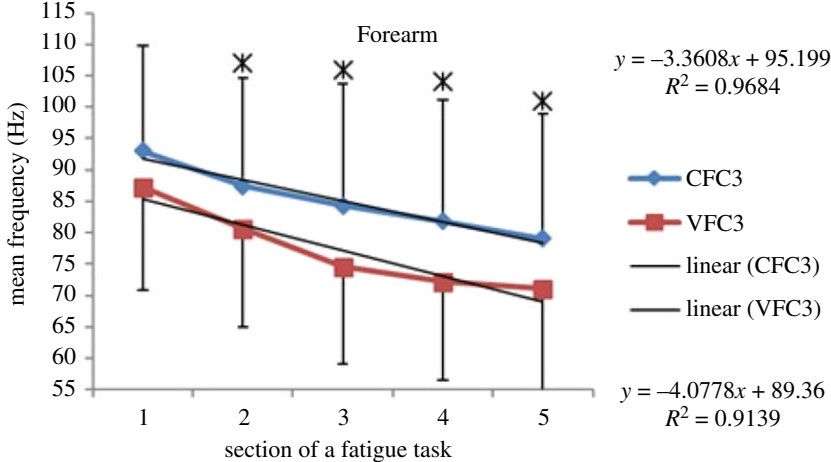

**Figure 12.** MEF values for the Forearm for the five consecutive sections of the fatigue effort during the progression of the third fatiguing exercise effort performed, under control and vibration condition, CFC3 versus VCF3. *Y*-values represent slopes of the best linear fit for C (top) and V (bottom) conditions quantified with least-squares method.

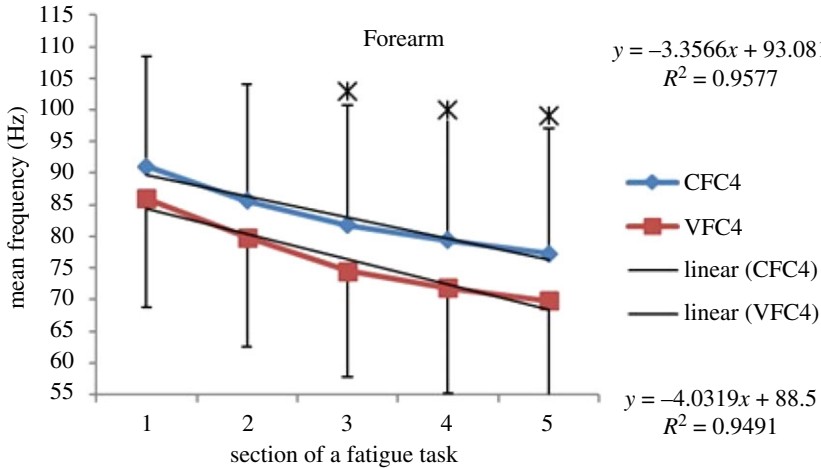

**Figure 13.** MEF values for the Forearm for the five consecutive sections of the fatigue effort during the progression of the fourth fatiguing exercise effort performed, under control and vibration condition, CFC4 versus VCF4. *Y*-values represent slopes of the best linear fit for C (top) and V (bottom) conditions quantified with least-squares method.

versus VFC1). Also, participants were able to sustain the fatiguing contractions for less required times under the vibration compared to the control (table 1). Only Triceps and Forearm showed statistically significant lower MEF values under vibration conditions as compared to their respective control conditions (figures 6–13). As the exercise progressed to the next consecutive effort, i.e. from CFC1 to CFC2 and so on, gradual decrease in the overall MEF can be observed irrespective of the (vibration/control) treatment condition and the muscle group (figures 6–13). For example, average MEFs in the second contraction effort (CFC2) are lower than the first (CFC1), likely an effect and indication of muscle fatigue.

## 3.4. EMG amplitude results—during fatigue exercise—EMGrms

Within each fatiguing effort, i.e. CFC or VFC, as the exercise/treatment progressed in time, the EMGrms amplitude trend under vibration conditions separated more from the control conditions. EMG amplitude under vibration condition efforts increased with a steeper positive slope indicating higher neuromuscular activity compared with the corresponding control effort (refer to electronic supplementary material, figures S14–S25). As the consecutive fatiguing efforts progressed the separation between the vibration and control EMG amplitudes mostly remained at the same level (electronic supplementary material, figures S14–S25). Thus, the EMGrms amplitude values under vibration condition were higher

(with the exception of Triceps CFC4 versus VFC4) compared with their control counterpart despite the progression of fatigue tasks from CFC1/VFC1 to CFC4/VFC4. This suggests that as the exercise efforts progressed, the vibration stimulation combined with isometric contraction continued to induce higher neuromuscular activity leading to increased EMG amplitude levels in the engaged muscles compared to the control exercises. However, only the Forearm showed statistically significant ($p < 0.05$) higher EMGrms amplitude values under vibration conditions compared with the respective control conditions (electronic supplementary material, figures S22–S25). Nevertheless, all three muscles showed trends of EMG amplitude separation described above with the Forearm showing the largest difference between the vibration and the control treatment EMGrms values followed by the Biceps and Triceps.

# 4. Discussion

## 4.1. Overall fatiguing effects

The effectiveness and the ability of vibration exercise(s) to induce enhanced neuromuscular activity have been investigated primarily through EMG amplitude responses. Only a small number of studies have investigated the fatiguing effects of vibration stimulation through the analysis of EMG frequency variables. Mischi *et al*. [20] and Xu *et al*. [35] have recently investigated the fatiguing effects of vibration through sEMG frequency analysis. Our vibration stimulation device and study design differ from these earlier reported studies [20,35].

The results of this ULV study confirm that in comparison with the near-maximal (80% of MVC) isometric contraction alone, near-maximal (80% of MVC) isometric contraction superimposed on vibration stimulation (of 30 Hz–0.4 mm) elicits an increased neuromuscular response in the upper limbs. This leads to a higher fatigue in the engaged muscles during the vibration treatment, as judged through decreased capacity to maintain specific force levels. Equally importantly, the results also indicate that muscles display a higher degree of myoelectric manifestation of fatigue during vibration treatment efforts compared to the control. The results also imply potential alterations in neuromuscular mechanisms to cope with the higher neuromuscular load/demand induced by the vibration superimposed on isometric contraction.

Although all three muscles investigated display higher fatigue and enhanced neuromuscular activity, the effect of vibration stimulation is different on different muscles. It was anticipated that due to their role in flexion, the agonist muscles (Biceps and Forearm) would exhibit higher fatigue and neuromuscular activity than the antagonist (Triceps). The results indeed confirm that the Biceps and Forearm display higher neuromuscular activity (EMGrms) during vibration treatment/fatigue efforts over the Tricep muscles. However, interestingly the Triceps also displayed more significant ($p < 0.05$) reduction in MEF, hence higher fatigue levels compared with the Biceps during the vibration treatment/fatigue efforts. Co-activation of the agonist and antagonist has been reported to maintain the joint angle/stability during the vibration stimulation in the upper limbs [36]. It seems that although the Tricep is not the prime mover in the force producing flexion task, its role in maintaining the elbow joint stability during the stimulation has an effect on its neuromuscular activity leading to lower MEF, hence higher fatigue.

The difference in the neuromuscular response of the muscles could be attributed partially to their proximity to the vibration source [37]. A higher vibration transmission can lead to comparably higher neuromuscular activity over the corresponding control treatment [37]. The Forearm was the most proximal muscle group to the vibration actuator being investigated here and shows the most significant ($p < 0.05$) higher neuromuscular response (EMGrms) and the most significant ($p < 0.05$) fatigue manifestation (MEF and MDF) compared to the control condition during the vibration treatment/fatigue tasks.

The difference in the neuromuscular response of the muscles to the vibration stimulation also depends on the muscle pre-stretch, i.e. muscle length/contraction prior to the stimulation along with the vibration characteristics [15,17]. This might partially explain why despite showing relatively higher neuromuscular activity and fatigue manifestation during the vibration treatment/efforts (figures 11–13), the Forearm did not display a similar level of fatigue manifestation post treatment as the Biceps (figure 4 and table 2). The Biceps displayed the most significant ($p < 0.05$) different EMGrms values post vibration treatment (MVC3 and MVC4) compared to the control (figure 5 and table 3). The higher level of fatigue manifestation in Biceps post vibration treatment compared to the Forearm could be also related to the higher level of pre-contraction the Biceps had to produce in order to facilitate the elbow flexion task.

**Table 2.** Two-way ANOVA results comparing MEF means between effort levels (MVC1 to MVC4) and between control and vibration.

| intervention and muscle group | effect of treatment—MVC1 to MVC4 (significant difference between effort levels, $p$-value) | effect of treatment V or C condition (significant difference between V and C condition, $p$-value) |
|---|---|---|
| Biceps | yes, $p = 0.018$ | no, $p = 0.099$ |
| Forearm | yes, $p = 0.005$ | no, $p = 0.889$ |
| Triceps | yes, $p = 0.002$ | no, $p = 0.230$ |

**Table 3.** Two-way ANOVA results comparing EMGrms means between effort levels (MVC1–MVC4) and between control and vibration.

| intervention and muscle group | effect of treatment—MVC1 to MVC4 (significant difference between effort levels, $p$-value) | effect of treatment V or C condition (significant difference between V and C condition, $p$-value) |
|---|---|---|
| Biceps | yes, $p = 0.019$ | yes, $p = 0.045$ |
| Forearm | yes, $p = 0.033$ | no, $p = 0.300$ |
| Triceps | yes, $p = 0.002$ | yes, $p = 0.006$ |

The results strongly indicate that vibration stimulation due to its higher neuromuscular demand potentially alters the underlying fatigue mechanisms. These potential fatigue mechanisms are discussed in the section below.

## 4.2. Potential mechanisms leading to increased neuromuscular activity and fatigue

It has been reported that myoelectric manifestations of fatigue are induced by the two main physiological factors: peripheral (muscle) fatigue and central fatigue [23]. The peripheral/muscle fatigue relates to the reduction of conduction velocity of MU action potential and the central fatigue is characterized by the synchronization of MUs by the CNS to increase the mechanical output when the MU pool is recruited [23,30]. Based on the above relation, it has been suggested that the conduction velocity, MEF values of the sEMG are strongly indicative of the peripheral fatigue [22].

With regards to the exercises in this study, during isometric constant force fatiguing contractions, conduction velocity seems to be particularly correlated with MEF of the sEMG [38]. Thus, one can argue, MEF results in this study can reliably reflect CV and hence the peripheral fatigue.

Further, it is well established that during sustained contractions the PSD of the sEMG signal compresses and shifts towards the lower frequencies leading to a reduction in MEF [39,40]. MEF values in this study have shown clear and consistent reduction as the fatiguing efforts progress and the MEF under vibration have shown higher reduction in MEF, which is indicative of reducing CV in active motor units. Further, decrease in CV is related to increase in fatigue [39,40]. Based on the higher level of reduction in MEF values compared to the control, the results of this study strongly indicate that vibration superimposed on isometric contraction induces higher fatigue compared with the control. This was true of all the muscles tested in this study, that is agonist (Biceps and Forearm) and antagonist (Triceps). Further, all the above muscles have also shown higher EMGrms values, corresponding to their lower MEF values, in almost all the vibration conditions compared to the control. Increase in EMGrms has been associated with increase in fatigue under sustained contractions [41]. This further affirms the higher fatiguing effects of vibration exercise superimposed with isometric contractions.

It is important to note that the vibration superimposed with isometric contractions continued to induce a higher rate of fatigue as the exercise progressed to the subsequent efforts, thus accentuating the overall fatiguing effect in the engaged muscles compared to their respective control conditions.

The enhanced neuromuscular activity observed during vibration stimulation has been also ascribed to the tonic vibration reflex (TVR). Further, although not in indirect vibration settings (i.e. WBV/ULV), however in direct vibration stimulation of upper limbs, increase in muscle fatigue has been attributed to

the TVR [42]. It has been reported that MU synchronization has a direct effect on the TVR and hence the muscle fatigue [42,43]. Thus, the MU synchronization and hence the central fatigue mechanisms could play a role in the responses to vibration stimulation superimposed with isometric contractions, although the exact mechanisms of MU synchronization and central fatigue under indirect vibration stimulations are still unclear. Given that vibration stimulation can change the central motor commands excitability, CNS's role in influencing responses to vibration superimposed with isometric contractions cannot be excluded [44].

Further, it is important to note that, during MVC immediately post vibration treatment (i.e. MVC3s) all the muscles displayed significantly higher MEF values respective to their controls and higher values for MVC after the recovery (i.e. MVC4s). This is puzzling and one would expect the opposite, i.e. lower MEF values post vibration treatment. This is despite their corresponding EMGrms values being lower, hence indicating higher mechanical fatigue post vibration treatment compared to the control. The increased MEF values could be due to potentiation of the stretch reflex observed following the indirect vibration exercise, which leads to the recruitment of high threshold MUs leading to the increase in MEF [45].

To summarize, the above results confirm that vibration superimposed on isometric contraction not only leads to higher fatigue levels in the engaged muscles but also alters the neuromuscular function immediately post treatment.

## 4.3. Limitations of the study

The vibration characteristics of this study were based on previous evidence which indicated that 30 Hz frequency stimulation combined with near-maximal (greater than 70% MVC) isometric contraction could elicit the largest neuromuscular response compared with control condition in the upper limbs [20]. Based on this evidence, the experiments in this study were limited only to the specific variables of excitation, i.e. 30 Hz–0.4 mm vibration, 80% of MVC force level, Biceps, Forearm and Triceps muscles and elbow (isometric) flexion task. Hence, the fatiguing effects of this stimulation and conclusions drawn from this study may not be extrapolated to other vibration characteristics, contraction types and levels and muscle groups [46–48].

In this study, no filtering was performed to remove the so-called 'motion artefacts' from the sEMG data obtained. This might be considered a limitation. However, it is important to note that a recent ULV study found no differences in the MEF and RMS values after removing the so-called motion artefacts [35]. Further, other studies have reported that the so-called 'motion artefacts' are in fact 'stretch reflex' responses and hence filtering them might actually remove important information about stretch reflex responses and central fatigue mechanisms [33].

## 5. Conclusion

(A) Near-maximal (80% of MVC) isometric fatiguing contractions superimposed on vibration stimulation leads to a higher rate of fatigue development compared to the isometric contraction alone in the upper limb muscles.

(B) The higher rates of fatigue observed under vibration in all the muscles studied are associated with correspondingly higher rates of neuromuscular activations.

(C) Vibration superimposed on isometric contraction continues to induce higher fatigue levels in the subsequent fatiguing efforts, thus accentuating the overall fatiguing effect on the engaged muscles.

(D) Post vibration treatment MVCs show higher manifestation of mechanical fatigue compared with the control. However, these post vibration MVCs also display higher levels of MEF (and MDF) implying alteration in neuromuscular function post treatment.

(E) Vibration superimposed on isometric contraction not only seems to alter the neuromuscular function during fatiguing efforts by inducing higher neuromuscular load but also post vibration treatment, potentially through the augmentation of stretch reflex and/or higher central motor command excitability.

(F) Both peripheral and central fatigue mechanisms are likely to play a role in the alteration of neuromuscular function observed, however, the role of peripheral fatigue mechanisms may be more prominent.

(G) The observed increase in fatigue and neuromuscular activity under vibration conditions is evident not only in agonist but also antagonist muscles, although the effect of vibration seem to depend on the muscle role (agonist/antagonist), muscle pre-stretch length, contraction level, vicinity to the vibration source and the vibration parameters.

Ethics. Experiments were approved by the College Ethics Review Board (CERB) of the College of Life Sciences and Medicine, University of Aberdeen, Aberdeen, Scotland, UK. Reference CERB/2011/3/573.

Data accessibility. Data available from the Dryad Digital Repository: https://doi.org/10.5061/dryad.jq8bg5c [49].

Authors' contributions. M.C., R.D.N. and A.N.P. conceptualized and designed the study. A.N.P. acquired the data and processed and analysed the data from the study. A.N.P. interpreted results from the data and R.D.N. and M.C. provided critical input to the results. A.N.P. drafted the original manuscript and finalized the manuscript. All authors read and revised the manuscript critically for important intellectual content, and approved the final manuscript for publication

Competing interests. The authors declare no competing interests.

Funding. Authors thank Scottish Funding Council (SFC) for the North East of Scotland Technology (NESTech) Seed Fund to support the study.

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
