## [Reviewer comments · Royal Society Open Science]

Review History

RSOS-190019.R0 (Original submission)

Review form: Reviewer 1 (Riccardo Di Giminiani)

Is the manuscript scientifically sound in its present form?

Yes

Are the interpretations and conclusions justified by the results?

Yes

Is the language acceptable?

Yes

Is it clear how to access all supporting data?

Yes

Do you have any ethical concerns with this paper?

No

Have you any concerns about statistical analyses in this paper?

No

Recommendation?

Accept with minor revision (please list in comments)

Comments to the Author(s)**Significance**

The authors investigate an interesting topic regarding the neuromuscular response in the upper limb muscles by superimposing indirect localized vibration to a defined level of isometric muscle contraction (by using a new device).

Findings of this study suggest that near maximal isometric fatiguing contractions superimposed on vibration stimulation lead to a higher rate of fatigue development compared to the isometric contraction alone. Vibration superimposed on isometric contraction alters the neuromuscular function during fatiguing efforts and post vibration treatment, thus the results contribute to increase the body of knowledge.

Accuracy of title and abstract

Title and abstract of the manuscript are adequate and provide an accurate understating of inquiry of the study.

Clarity of hypothesis and rationale

Hypothesis should be more specific and underline the novelty of the present study; that is: the level of isometric muscle contraction is exactly defined by means of a new pulley system!

Actually, the two hypothesis appear too general.

Adequacy of experimental design and methods

This section is clear and well detailed.

Quality of data and presentation of results

In the section results should be reported the p-values and the effect size.

Tables and figures should be reorganized. In my opinion, the use of tables and figures to show the same results is not appropriate.

The symbols indicating the significant differences should be included in the figures. In this way the reader is facilitated to understand the section results.

Considering the large standard deviations, the Authors could report the standard errors in place of standard deviations.

Please, check the symbols (indicating the statistical significance) in the table 3 and 4.

Length and appropriateness of discussion

This section opens with the restated aim of the study which is consistent with the purposes expressed in the introduction. The physiological reasoning has been expanded and the limitations addressed.

Page 18, lines 42-44. Please, include some references.

Please, check the reference 42.

In synthesis, references reflect the most relevant and recent articles in the area of the study.

Review form: Reviewer 2

Is the manuscript scientifically sound in its present form?

Yes

Are the interpretations and conclusions justified by the results?

No

Is the language acceptable?

No

Is it clear how to access all supporting data?

Not Applicable

Do you have any ethical concerns with this paper?

No

Have you any concerns about statistical analyses in this paper?

No

Recommendation?

Major revision is needed (please make suggestions in comments)

Comments to the Author(s)

In the current manuscript, authors have tried to understand the mechanism of neuromuscular fatigue for upper limb during isometric exercise with and without vibration stimulation. The topic of research is worthwhile and would increase our current understanding of vibration stimulation at a higher tension level of muscles during the task for the upper limb. In general, the study is methodology is explained well but the presentation of results and stats in its current form fails to succinctly represent the outcome, It needs rewriting.

I would like that author to address/consider the following points to improve the presentation of the paper

- 1) Please cite some literature which talks about the decrease in MEF as a measure of fatigue in the introduction.
- 2) Result section has a lot of emphasis on subjective/trends, which can be just part of just normal variance when you record measure like MVC, please only present the significant results mainly and trends can be mentioned in very brief and that too in discussion not in the results section.
- 3) There are too many figures, maybe authors can combine them in subplots to improve readability, also use the asterisk, where necessary to show significant differences as authors have done in later figures
 - Maybe an author can consider presenting %changes
- 4) Please avoid the explanation of the results in the results section like following lines,
page 7 lines 36-41,45-48,51-53
page 9 lines 42-43
they need to be moved in the discussion section.
- 5) section "EMG Amplitude Results- Pre and Post Fatigue Exercise- EMGrms" needs to be completely rewritten after reading it number of times, it's not clear what's happening, again please do not overemphasize on trends and focus on statistically significant results
- 5) Table 4 have P values <0.05 but they are not marked as significant, please check and explain
- 6) in "EMG Frequency Results- During Fatigue Exercise- Mean Frequency (MEF)" section authors are talking about slopes, but not an objective measure of the slope is discussed in methods section or even in results, authors are merely describing the trends visible from figures. Authors can

present % change in slopes. Again, please refrain from the explanation of results page 10 line 53-56, line 57-60. Page 12 line 1-11. In this section the number of figures can be reduced, authors can consider using subplots.

7) It's hard to conclude anything from the presented results a lot more focus is required in presenting the results. Because results are obscure so its hard to read the discussion section. Once the Results section is revamped based on above comments, hopefully, it will easier to read the discussion and make sense of the work presented.

8) Journal name is missing from the following citation, please double check all the references -Mischi M, Cardinale M Muscle electrical activity during force modulation exercise. vol. 2008 pp. 2065-2068.

Review form: Reviewer 3

Is the manuscript scientifically sound in its present form?

Yes

Are the interpretations and conclusions justified by the results?

Yes

Is the language acceptable?

Yes

Is it clear how to access all supporting data?

Yes

Do you have any ethical concerns with this paper?

No

Have you any concerns about statistical analyses in this paper?

Yes

Recommendation?

Accept with minor revision (please list in comments)

Comments to the Author(s)

The manuscript presents an interesting study on the application of vibration exercise to the upper-limb muscles in order to evaluate fatigue. The results show, as expected, increased myoelectric fatigue with vibration exercise, with the exception of an unexpected increase in the EMG mean frequency during the MVC test post exercise. The manuscript is clear and well written.

Hereafter I is the list of my comments and remarks.

Page 4: the concept of "indirect vibration", as opposed to direct vibration, should be explained in the introduction as not all readers are familiar with this terminology.

Page 4, line 54: I cannot find in the manuscript results "post recovery" but only before and after the exercises. Please explain.

Page 7: it would be useful to the reader to have a figure showing the exact electrode positioning, with respect to the fiber orientation as well as to the muscle innervation.

Page 8, EMG data processing: have the authors applied a band-pass filter (e.g., 20-400 Hz) before the analysis? Low-frequency noise (<20 Hz) could explain the increase in MEF obtained after vibration exercise.

Page 9, line 2: Additional evidence supporting the authors' reasoning on motion artifacts is provided by the following reference:

Xu, L., Negro, F., Xu, Y., Rabotti, C., Schep, G., Farina, D., & Mischi, M. (2018). Does vibration superimposed on low-level isometric contraction alter motor unit recruitment strategy?. *Journal of neural engineering*, 15(6), 066001.

Page 9, Statistical analysis: It is unclear how the different segments of each recording are compared in the statistics since they are all of different length and timing, depending on the subject and the trial (which stops at 60% of the MVC). This also refers to the plots in Figs 10-17. It seems that the values from measurements 1 to 5 are not necessarily referring to the same time. The authors should elaborate on this.

Page 9, Statistical analysis: is the use of a t-test motivated by a Gaussian distribution of the data?

In the fatigue plots, Figs 10-17, the authors may also like to consider the decay (fatiguing rate), taken as the angular coefficient of the linear fit to the points. This should differ for vibration and control.

Decision letter (RSOS-190019.R0)

19-Jun-2019

Dear Dr Pujari,

The editors assigned to your paper ("Fatiguing Effects of Indirect Vibration Stimulation in Upper Limb Muscles- post and during Isometric Contractions Superimposed on Upper Limb Vibration") have now received comments from reviewers. We would like you to revise your paper in accordance with the referee and Associate Editor suggestions which can be found below (not including confidential reports to the Editor). Please note this decision does not guarantee eventual acceptance.

Please submit a copy of your revised paper before 12-Jul-2019. Please note that the revision deadline will expire at 00.00am on this date. If we do not hear from you within this time then it will be assumed that the paper has been withdrawn. In exceptional circumstances, extensions may be possible if agreed with the Editorial Office in advance. We do not allow multiple rounds of revision so we urge you to make every effort to fully address all of the comments at this stage. If deemed necessary by the Editors, your manuscript will be sent back to one or more of the original reviewers for assessment. If the original reviewers are not available, we may invite new reviewers.

To revise your manuscript, log into <http://mc.manuscriptcentral.com/rsos> and enter your Author Centre, where you will find your manuscript title listed under "Manuscripts with Decisions." Under "Actions," click on "Create a Revision." Your manuscript number has been

appended to denote a revision. Revise your manuscript and upload a new version through your Author Centre.

- Data accessibility

If you wish to submit your supporting data or code to Dryad (<http://datadryad.org/>), or modify your current submission to dryad, please use the following link:
<http://datadryad.org/submit?journalID=RSOS&manu=RSOS-190019>

- Competing interests

- Authors' contributions

- Acknowledgements

- Funding statement

Kind regards,

Alice Power

Editorial Coordinator

on behalf of Dr Monica Daley (Associate Editor) and Kevin Padian (Subject Editor)

Associate Editor's comments (Dr Monica Daley):

The expert reviewers have made a number of suggestions for revising this paper to more clearly convey the specific reasoning and hypotheses of the study, and to more clearly and concisely present the results. Addressing these comments will require substantial revision to the paper, and possibly another round a review.

When making the revisions, please particularly focus on making the message of the paper specific, focused and clear. Please try to consolidate the figures and results to focus on presenting the data that is directly needed. Seventeen figures seems like too many, and the data are currently presented in a format that wastes a lot of ink and page space. For example, results for the same muscle across conditions might be consolidated into a single multi-panel figure with unnecessary gridlines and box outlines removed. Consider organizing and consolidating the figures in a way that most directly conveys the key findings required test the main hypotheses of the paper.

Comments to Author:

Reviewers' Comments to Author:

Reviewer: 1

Comments to the Author(s)

Significance

The authors investigate an interesting topic regarding the neuromuscular response in the upper limb muscles by superimposing indirect localized vibration to a defined level of isometric muscle contraction (by using a new device).

Findings of this study suggest that near maximal isometric fatiguing contractions superimposed on vibration stimulation lead to a higher rate of fatigue development compared to the isometric contraction alone. Vibration superimposed on isometric contraction alters the neuromuscular

function during fatiguing efforts and post vibration treatment, thus the results contribute to increase the body of knowledge.

Accuracy of title and abstract

Title and abstract of the manuscript are adequate and provide an accurate understating of inquiry of the study.

Clarity of hypothesis and rationale

Hypothesis should be more specific and underline the novelty of the present study; that is: the level of isometric muscle contraction is exactly defined by means of a new pulley system! Actually, the two hypothesis appear too general.

Adequacy of experimental design and methods

This section is clear and well detailed.

Quality of data and presentation of results

In the section results should be reported the p-values and the effect size.

Tables and figures should be reorganized. In my opinion, the use of tables and figures to show the same results is not appropriate.

The symbols indicating the significant differences should be included in the figures. In this way the reader is facilitated to understand the section results.

Considering the large standard deviations, the Authors could report the standard errors in place of standard deviations.

Please, check the symbols (indicating the statistical significance) in the table 3 and 4.

Length and appropriateness of discussion

This section opens with the restated aim of the study which is consistent with the purposes expressed in the introduction. The physiological reasoning has been expanded and the limitations addressed.

Page 18, lines 42-44. Please, include some references.

Please, check the reference 42.

In synthesis, references reflect the most relevant and recent articles in the area of the study.

Reviewer: 2

Comments to the Author(s)

In the current manuscript, authors have tried to understand the mechanism of neuromuscular fatigue for upper limb during isometric exercise with and without vibration stimulation. The topic of research is worthwhile and would increase our current understanding of vibration stimulation at a higher tension level of muscles during the task for the upper limb. In general, the study is methodology is explained well but the presentation of results and stats in its current form fails to succinctly represent the outcome, It needs rewriting.

I would like that author to address/consider the following points to improve the presentation of the paper

- 1) Please cite some literature which talks about the decrease in MEF as a measure of fatigue in the introduction.
- 2) Result section has a lot of emphasis on subjective/trends, which can be just part of just normal variance when you record measure like MVC, please only present the significant results mainly and trends can be mentioned in very brief and that too in discussion not in the results section.

3) There are too many figures, maybe authors can combine them in subplots to improve readability, also use the asterisk, where necessary to show significant differences as authors have done in later figures

-Maybe an author can consider presenting %changes

4) Please avoid the explanation of the results in the results section like following lines, page 7 lines 36-41,45-48,51-53

page 9 lines 42-43

they need to be moved in the discussion section.

5) section "EMG Amplitude Results- Pre and Post Fatigue Exercise- EMGrms" needs to be completely rewritten after reading it number of times, it's not clear what's happening, again please do not overemphasize on trends and focus on statistically significant results

5) Table 4 have P values <0.05 but they are not marked as significant, please check and explain

6) in "EMG Frequency Results- During Fatigue Exercise- Mean Frequency (MEF)" section authors are talking about slopes, but not an objective measure of the slope is discussed in methods section or even in results, authors are merely describing the trends visible from figures. Authors can present % change in slopes. Again, please refrain from the explanation of results page 10 line 53-56, line 57-60. Page 12 line 1-11. In this section the number of figures can be reduced, authors can consider using subplots.

7) It's hard to conclude anything from the presented results a lot more focus is required in presenting the results. Because results are obscure so its hard to read the discussion section. Once the Results section is revamped based on above comments, hopefully, it will easier to read the discussion and make sense of the work presented.

8) Journal name is missing from the following citation, please double check all the references -Mischi M, Cardinale M Muscle electrical activity during force modulation exercise. vol. 2008 pp. 2065-2068.

Reviewer: 3

Comments to the Author(s)

The manuscript presents an interesting study on the application of vibration exercise to the upper-limb muscles in order to evaluate fatigue. The results show, as expected, increased myoelectric fatigue with vibration exercise, with the exception of an unexpected increase in the EMG mean frequency during the MVC test post exercise. The manuscript is clear and well written.

Hereafter I is the list of my comments and remarks.

Page 4: the concept of "indirect vibration", as opposed to direct vibration, should be explained in the introduction as not all readers are familiar with this terminology.

Page 4, line 54: I cannot find in the manuscript results "post recovery" but only before and after the exercises. Please explain.

Page 7: it would be useful to the reader to have a figure showing the exact electrode positioning, with respect to the fiber orientation as well as to the muscle innervation.

Page 8, EMG data processing: have the authors applied a band-pass filter (e.g., 20-400 Hz) before the analysis? Low-frequency noise (<20 Hz) could explain the increase in MEF obtained after vibration exercise.

Page 9, line 2: Additional evidence supporting the authors' reasoning on motion artifacts is provided by the following reference:

Xu, L., Negro, F., Xu, Y., Rabotti, C., Schep, G., Farina, D., & Mischi, M. (2018). Does vibration superimposed on low-level isometric contraction alter motor unit recruitment strategy?. *Journal of neural engineering*, 15(6), 066001.

Page 9, Statistical analysis: It is unclear how the different segments of each recording are compared in the statistics since they are all of different length and timing, depending on the subject and the trial (which stops at 60% of the MVC). This also refers to the plots in Figs 10-17. It seems that the values from measurements 1 to 5 are not necessarily referring to the same time. The authors should elaborate on this.

Page 9, Statistical analysis: is the use of a t-test motivated by a Gaussian distribution of the data?

In the fatigue plots, Figs 10-17, the authors may also like to consider the decay (fatiguing rate), taken as the angular coefficient of the linear fit to the points. This should differ for vibration and control.

Author's Response to Decision Letter for (RSOS-190019.R0)

See Appendix A.

RSOS-190019.R1 (Revision)

Review form: Reviewer 1 (Riccardo Di Giminiani)

Is the manuscript scientifically sound in its present form?

Yes

Are the interpretations and conclusions justified by the results?

Yes

Is the language acceptable?

Yes

Do you have any ethical concerns with this paper?

No

Have you any concerns about statistical analyses in this paper?

No

Recommendation?

Accept as is

Comments to the Author(s)

Thanks for a thorough revision of the manuscript. In my opinion the text has improved considerably and it is ready for publication.

Review form: Reviewer 2**Is the manuscript scientifically sound in its present form?**

Yes

Are the interpretations and conclusions justified by the results?

Yes

Is the language acceptable?

Yes

Do you have any ethical concerns with this paper?

No

Have you any concerns about statistical analyses in this paper?

No

Recommendation?

Accept as is

Comments to the Author(s)

Thanks for the revised version. I am satisfied with the replies.

Decision letter (RSOS-190019.R1)

08-Sep-2019

Dear Dr Pujari,

I am pleased to inform you that your manuscript entitled "Fatiguing Effects of Indirect Vibration Stimulation in Upper Limb Muscles- post and during Isometric Contractions Superimposed on Upper Limb Vibration" is now accepted for publication in Royal Society Open Science.

on behalf of Dr Monica Daley (Associate Editor) and Kevin Padian (Subject Editor)
openscience@royalsociety.org

Editor comments:

Thanks for your revision and congratulations. The reviewers are satisfied with your revision and we are happy to accept it for publication.

Reviewer comments to Author:

Reviewer: 2

Comments to the Author(s)

Thanks for the revised version. I am satisfied with the replies.

Reviewer: 1

Comments to the Author(s)

Thanks for a thorough revision of the manuscript. In my opinion the text has improved considerably and it is ready for publication.

Follow Royal Society Publishing on Twitter: [@RSocPublishing](https://twitter.com/RSocPublishing)

Appendix A

Authors thank all the referees and associate editor for their positive as well as critical comments about the paper. Their detailed comments and suggestions has improved the manuscript significantly. Below we address all the reviewer's comments and have carried out relevant changes in the manuscript (these changes are highlighted in red in the manuscript).

Response to associate editors' comments: Hypothesis has been rewritten to make it more focussed and specific. Figures have been redone with total number of figures coming to 13 from 17. Presentation of the data in the Figures and Tables has been improved to convey the results and message more clearly and succinctly. Results and discussion sections has been tightened to address the reviewers comments and to convey message more clearly.

Reviewer 1

Comment: Hypothesis should be more specific and underline the novelty of the present study; that is: the level of isometric muscle contraction is exactly defined by means of a new pulley system!

Actually, the two hypotheses appear too general.

Response: Hypothesis has been amended to make it more specific to reflect how muscle contraction is characterized by our new device.

Comment: In the section results should be reported the p-values and the effect size.

Response: It has been ensured that all the results report corresponding p- values, the text has been updated to reflect this. To address your effect size suggestion, we have now carried out further statistical analysis. To analyse the effect of treatment (Control Vs Vibration) and effort (MVC1 to MVC4), we have employed 2 way Anova and the results comparing EMGrms means between effort levels (MVC1 to MVC4) and between control and vibration are now included in Table 1 and 2 in the manuscript.

Comment: Tables and figures should be reorganized. In my opinion, the use of tables and figures to show the same results is not appropriate.

Response: Figures are reorganized now. Especially, multiple figures have been combined to reduce figure numbers and to present the data more succinctly. For example, figures 1 to 3 and 4 to 6 have been combined into new Figures 1 and 2 respectively, combining 3 muscle groups in 1 Figure. Tables depicting p-values have been removed instead Figures have been improved to convey statistically significant differences.

Comment: The symbols indicating the significant differences should be included in the figures. In this way the reader is facilitated to understand the section results.

Response: Completely agree and thanks for pointing this out. It was an oversight not to include symbols indicating the significant differences. They have now been included on all the figures.

Comment: Considering the large standard deviations, the Authors could report the standard errors in place of standard deviations.

Response: Agree, Figures are now reported with standard error instead of standard deviation.

Comment: Please, check the symbols (indicating the statistical significance) in the table 3 and 4.

Response: Thanks for pointing this out, these tables have been deleted as per your earlier suggestion, to not to repeat the information in Figures and Tables. Instead, new tables representing Anova results are included.

Comment: Page 18, lines 42-44. Please, include some references.

Response: References have been included as suggested to support the statement.

Comment: Please, check the reference 42.

Response: Reference has been updated.

Reviewer 2:

Comment 1: Please cite some literature which talks about the decrease in MEF as a measure of fatigue in the introduction.

Response: Literature citing MEF as a representation of myoelectric manifestation of fatigue has now been included in the introduction section along with brief reasoning behind this choice. This text can be found just before hypothesis.

Comment 2: Result section has a lot of emphasis on subjective/trends, which can be just part of just normal variance when you record measure like MVC, please only present the significant results mainly and trends can be mentioned in very brief and that too in discussion not in the results section.

Response: This is a really useful comment to ensure that manuscript relies on most objective elements when discussing and positing. Results section has now been updated with less emphasis on subjective trends and any mentions of these trends have been moved to discussion section.

Comment 3: There are too many figures, maybe authors can combine them in subplots to improve readability, also use the asterisk, where necessary to show significant differences as authors have done in later figures.

Response: Figures have been combined to improve readability. For example, figures 1 to 3 and 4 to 6 have been combined into new Figures 1 and 2 respectively, combining 3 muscle groups in 1 Figure. It has been ensured that all the figures display significant differences marked with asterisk sign.

Comment 4: Please avoid the explanation of the results in the results section like following lines, page 7 lines 36-41, 45-48, 51-53 page 9 lines 42-43 they need to be moved in the discussion section.

Response: Results section has now been updated to remove explanation of the results from this section to discussion.

Comment 5: Section "EMG Amplitude Results- Pre and Post Fatigue Exercise- EMGrms" needs to be completely rewritten after reading it number of times, it's not clear what's happening, again please do not overemphasize on trends and focus on statistically significant results

Response: This section has been rewritten to convey the message in clearer terms without overemphasizing the trends and focussing on the significant results. Hopefully this reads well now.

Comment 6: Table 4 have P values <0.05 but they are not marked as significant, please check and explain.

Response: All the tables have now been re-checked for p values and any mistakes have been corrected. In addition, based on other reviewer's suggestion, tables depicting the P-values have been removed so not to repeat the same information in the Figures and Tables. Instead, Figures now clearly convey statistically significant differences with these values mentioned in brackets in results and discussion text where appropriate.

To analyse the effect of treatment (Control Vs Vibration) and effort (MVC1 to MVC4), we have now employed 2 way Anova and the results comparing EMGrms means between effort levels (MVC1 to MVC4) and between control and vibration are now included in Table 1 and 2 in the manuscript.

Comment 7: In EMG Frequency Results- During Fatigue Exercise- Mean Frequency (MEF)" section authors are talking about slopes, but not an objective measure of the slope is discussed in methods section or even in results, authors are merely describing the trends visible from figures. Authors can present % change in slopes. Again, please refrain from the explanation of results page 10 line 53-56, line 57-60. Page 12 line 1-11. In this section the number of figures can be reduced, authors can consider using subplots.

Response: Agree with this comment, an objective measure quantifying the slopes was not included. We have now quantified the reduction in MEF values i.e. changes in the slope with regression analysis (using least square method). Now all the fatigue figures include this quantified measure of change in slope, which now has been used to support our discussion. Explanation of results has been moved to discussion section. We have tried reducing the number of figures throughout the manuscript. Bar graphs in the results have now been combined to improve readability. However, due to the nature of the fatigue figures, combining them as subplots makes the figure illegible. Overlaying fatigue graphs is also not an option as the trends of the plots are similar; they make the graph lines obscure when overlaid on one another. Hence the fatigue figures have been kept similar however, each fatigue figure now includes quantified measure of the change of slope.

Comment 8: It's hard to conclude anything from the presented results a lot more focus is required in presenting the results. Because results are obscure so it's hard to read the discussion section. Once the Results section is revamped based on above comments, hopefully, it will easier to read the discussion and make sense of the work presented.

Response: Results section has now been improved to convey the message more clearly and hopefully reads better. To summaries, among others, following changes have been carried out to improve the focus and readability.

1) Explanation of results has been avoided in the results section and moved to discussion section. 2) Figures with bar graphs have been combined with statistical significant values clearly indicated. 3) Objective measure of slope change is now included to support the discussion. 4) Less emphasis is placed on trends.

Comment 9: Journal name is missing from the following citation, please double check all the references -Mischi M, Cardinale M Muscle electrical activity during force modulation exercise. vol. 2008 pp. 2065–2068.

Response: Reference has been corrected and all the references have been rechecked for their details.

Reviewer 3:

Comment Page 4: the concept of "indirect vibration", as opposed to direct vibration, should be explained in the introduction as not all readers are familiar with this terminology.

Response: Thanks for pointing this out. Explanation of direct and indirect vibration has been now included in the introduction.

Comment: Page 4, line 54: I cannot find in the manuscript results “post recovery” but only before and after the exercises. Please explain.

Response: Thanks for noticing this, the phrase should be post exercise and not post recovery. This has now been corrected in the manuscript.

Comment: Page 7: it would be useful to the reader to have a figure showing the exact electrode positioning, with respect to the fibre orientation as well as to the muscle innervation.

Response: This is good point; unfortunately, photographs of the electrodes placements were not taken at the time of experiments. However, further clarification about exact electrode placement used is provided on page 7 and reader is directed to SENIAM website for further details about the reference used for achieving precise electrode placements.

Comment: Page 8, EMG data processing: have the authors applied a band-pass filter (e.g., 20-400 Hz) before the analysis? Low-frequency noise (<20 Hz) could explain the increase in MEF obtained after vibration exercise.

Response: EMG data acquisition involved analogue band pass filtering (20-450Hz) at the point of acquisition. The text has now been update to clarify this on page 8.

Comment: Page 9, line 2: Additional evidence supporting the authors’ reasoning on motion artifacts is provided by the following reference:

Xu, L., Negro, F., Xu, Y., Rabotti, C., Schep, G., Farina, D., & Mischi, M. (2018). Does vibration superimposed on low-level isometric contraction alter motor unit recruitment strategy? *Journal of neural engineering*, 15(6), 066001.

Response: This is a more recent and relevant reference, this has now been included.

Comment: Page 9, Statistical analysis: It is unclear how the different segments of each recording are compared in the statistics since they are all of different length and timing, depending on the subject and the trial (which stops at 60% of the MVC). This also refers to the plots in Figs 10-17. It seems that the values from measurements 1 to 5 are not necessarily referring to the same time. The authors should elaborate on this.

Response: You are correct in pointing this out. Values from measurements 1 to 5 are actually with respect to time and they depend on the subject’s performance. Below explanation (which is already included in the ‘EMG Data processing of the Fatiguing Contractions’ section) should clarify your query to an extent. We have provided further explanation on how statistical analysis on these data sets was completed.

“To compare the relative fatigue levels attained by the individuals while performing the same relative efforts, the EMG data selection for the analysis began once the target force level of 80% of MVC was achieved and ended when the force level dropped down to 60% of MVC. Thus, the duration for which the EMG data was selected was precisely connected to an individual’s ability to sustain the target force. As expected, this period varied with individual and with the progression to the successive fatiguing contraction (Table 3).

The selection of EMG data was then divided into five equal sections (e.g. 60 seconds selection was divided into 5 sections of 12 seconds each). Then an average for each section was calculated to give a

single mean value for that section of the fatigue effort. Both the time domain (EMGrms) and frequency domain (mean (MEF) frequency) calculations were performed on the same selection of data.

Further statement included in the manuscript providing clearer explanation stating: “To carry out statistical analysis, each single mean value obtained for a specific section of the fatigue effort (1 to 5) for a single participant, was compared with the respective mean values of the other participants”.

Comment: Page 9, Statistical analysis: is the use of a t-test motivated by a Gaussian distribution of the data?

Response: Yes this is correct, choice of t-tests to establish the significance level was based on Gaussian distribution of the data. This is now clarified in the text in statistical analysis sub section.

Comment: In the fatigue plots, Figs 10-17, the authors may also like to consider the decay (fatiguing rate), taken as the angular coefficient of the linear fit to the points. This should differ for vibration and control.

Response: This is an excellent point, thanks for suggesting. We have now quantified the reduction in MEF values i.e. changes in the slope with regression analysis (using least square method). Now all the fatigue figures include this quantified measure of change in slope, which now has been used to support our discussion.